# Docker4Circ: A Framework for the Reproducible Characterization of circRNAs from RNA-Seq Data

**DOI:** 10.3390/ijms21010293

**Published:** 2019-12-31

**Authors:** Giulio Ferrero, Nicola Licheri, Lucia Coscujuela Tarrero, Carlo De Intinis, Valentina Miano, Raffaele Adolfo Calogero, Francesca Cordero, Michele De Bortoli, Marco Beccuti

**Affiliations:** 1Department of Computer Science, University of Turin, 10149 Turin, Italy; giulio.ferrero@unito.it (G.F.); nicola.licheri@unito.it (N.L.); carlo.deintinis@unito.it (C.D.I.); francesca.cordero@unito.it (F.C.); marco.beccuti@unito.it (M.B.); 2Department of Clinical and Biological Sciences, University of Turin, Orbassano, 10043 Turin, Italy; lucia.coscujuelatarrero@unito.it (L.C.T.); valentina.miano@gmail.com (V.M.); 3Center for Genomic Science, Italian Institute of Technology, 20139 Milan, Italy; 4Division of Cellular and Molecular Pathology, Department of Pathology, University of Cambridge, Addenbrooke’s Hospital, Cambridge CB2 0QQ, UK; 5Department of Molecular Biotechnology and Health Sciences, University of Turin, 10126 Turin, Italy; Raffaele.calogero@unito.it

**Keywords:** circRNA, reproducible analysis, pipeline, docker images

## Abstract

Recent improvements in cost-effectiveness of high-throughput technologies has allowed RNA sequencing of total transcriptomes suitable for evaluating the expression and regulation of circRNAs, a relatively novel class of transcript isoforms with suggested roles in transcriptional and post-transcriptional gene expression regulation, as well as their possible use as biomarkers, due to their deregulation in various human diseases. A limited number of integrated workflows exists for prediction, characterization, and differential expression analysis of circRNAs, none of them complying with computational reproducibility requirements. We developed Docker4Circ for the complete analysis of circRNAs from RNA-Seq data. Docker4Circ runs a comprehensive analysis of circRNAs in human and model organisms, including: circRNAs prediction; classification and annotation using six public databases; back-splice sequence reconstruction; internal alternative splicing of circularizing exons; alignment-free circRNAs quantification from RNA-Seq reads; and differential expression analysis. Docker4Circ makes circRNAs analysis easier and more accessible thanks to: (i) its R interface; (ii) encapsulation of computational tasks into docker images; (iii) user-friendly Java GUI Interface availability; and (iv) no need of advanced bash scripting skills for correct use. Furthermore, Docker4Circ ensures a reproducible analysis since all its tasks are embedded into a docker image following the guidelines provided by Reproducible Bioinformatics Project.

## 1. Introduction

CircRNAs are circular RNA molecules with no free 5’/3’ ends, formed by back-splicing (BS) events [1]. To date five categories of circRNAs have been defined, arising respectively from: (i) One or more exons of the linear transcript (exonic); (ii) an intron of the linear transcript (intronic); (iii) an antisense transcript including exons (antisense); (iv) a transcript of the same gene locus of the main linear transcript, but not exonic or intronic (intragenic); and (v) a transcript from nongenic regions (intergenic) [2]. Exonic circRNAs are the main category and here the mechanism is clearly dependent on the same machinery that carries out normal exon splicing, involving in this case a downstream 5’ splice site and an upstream 3’ splice site. In 2013, Jeck et al. described two possible models for circRNA formation [3]. In the “lariat driven circularization” model, an exon skipping event generates a long intron lariat containing the skipped exon(s), which undergoes back-splicing to generate a circRNA. The “intron-pairing-driven circularization” suggests that introns flanking the circularizing exon(s) are paired forming a stem-loop structure, due either to complementary sequences, such as inverted Alu, or to interacting RBPs bound to the two introns.

The expression of circRNAs has been reported in almost all tissues, with special enrichment in the brain, and displays features of tissue specificity that is often not correlated with the expression of the cognate linear isoforms. Interestingly, there are many reports of altered circRNA expression in pathological tissues, including cancer, and the relative stability of circRNAs due to the lack of free ends makes them detectable also in body fluids [4].

Several circRNAs databases are also becoming available including circBase [5], Tissue-Specific CircRNA Database (TSCD) [6], circRNADb [7], Circ2Disease [8], ExoRBase [9], Cancer Specific CircRNA Database (CSCD) [10], CircFunBase [11], and Circ2Traits [12], increasing the accessible information on annotated circRNAs.

Appreciation of circRNAs has escaped the common procedures for gene expression analysis until recently. However, aside from dedicated approaches such as Poly(A+)-depletion or RNAseR treatment followed by RNA-Seq, an increasing number of total RNA-Seq datasets suitable for circRNAs evaluation in a number of different biological contexts is nowadays available. This makes definitely relevant the possibility of studying circRNAs expression in a variety of cells and tissues using available datasets, thus expanding rapidly the knowledge on circRNAs regulation and functions in different experimental or pathological contexts.

The identification of circRNAs in RNA-Seq data relies solely on reads mapping to back-splicing junctions. The problem is analogous to identify novel splicing isoforms in linear transcripts, which may be quite hard when these isoforms are expressed at a low level, as indeed in the case of circRNAs. Once a back-splicing is predicted, establishing genomic features and reconstructing circRNA structure is necessary, based on the available data. Finally, differential expression analysis can be performed.

Many computational tools were developed for predicting circRNAs from RNA-Seq data [2,13], and different tools were proposed for the post-prediction analyses including FUCHS [14] and CIRI-AS [15] for defining circRNAs internal structures, Sailfish-circ [16], and circTest [17] for circRNAs quantification and differential expression analysis and CircView for visualization of circRNAs predictions [18]. Furthermore, our group recently proposed the CircHunter algorithm for the characterization and quantification of circRNAs using public RNA-Seq datasets [19]. All these aspects clearly highlight the need for workflows able to provide a comprehensive characterization of circRNAs. Extensive pipelines were designed for this purpose like CirCompara [20], Ularcirc [21], and circtools [22]; however, they do not meet the computational reproducibility standards suggested by Sandve and colleagues [23].

In this paper, we present Docker4Circ, a comprehensive framework for circRNAs analysis providing: (1) circRNAs prediction from RNA-Seq data; (2) classification and annotation of circRNAs over six public databases; (3) reconstruction of the back-splicing sequence; (4) internal alternative splicing for multi-exonic circRNAs; (5) alignment-free circRNAs quantification from RNA-Seq reads, and (6) differential expression analysis. The distinctive features of Docker4Circ are the usability and portability on all Unix-like systems achieved through docker containerization, an R interface and a Java Graphical User Interface (GUI); and the computational reproducibility of any performed analysis since it follows the guidelines provided by the Reproducible Bioinformatics Project (RBP) [24,25].

## 2. Results

### 2.1. A Framework to Create Modular Workflows for Reproducible Analysis of circRNA Data

We designed Docker4Circ as an integrated computational framework to the goal of providing all the common steps from RNA-Seq reads to full analysis of circRNA structure and expression, allowing a number of user-defined options in an easy interface. Docker4Circ consists of an R library (integrated into Docker4Seq package [23]) and a set of docker images. The framework includes public analysis tools for RNA-Seq read quality control (FASTQC), read alignment (BWA and STAR), circRNA prediction (CIRI2 and STARChip), circRNA internal sequence analysis (CIRI-AS), circRNA classification (CircHunter), and expression analysis (CircHunter and DESeq2). Furthermore, additional functions were included to allow the integration of multiple circRNAs predictions from different tools and the circRNAs annotation with the information stored in public databases. To facilitate the usage of these tools and to achieve the computational reproducibility of the analysis, all the Docker4Circ tools are pre-installed into a set of docker images (according to the guidelines of the RBP). The Docker4Seq R library provides a simplified user interface to run Docker4Circ, for which no knowledge of the docker commands is needed.

The framework functionalities are grouped in four modules: circRNAs prediction (Module 1), circRNAs classification and annotation (Module 2), the Back-Splicing (BS) sequence analysis (Module 3), and circRNAs expression analysis (Module 4) (Figure 1).

Moreover, to simplify the use of Docker4Circ for users with no scripting experience, R functions can be controlled by a dedicated GUI, which is a part the 4SeqGUI project https://github.com/mbeccuti/4SeqGUI (Figure 2a).

Details on the usage of each function are reported in the Appendix A of the manuscript. Docker4Circ functions and associated test data can be downloaded from https://github.com/kendomaniac/docker4seq.

#### 2.1.1. Module 1: circRNAs Prediction

This module is designed to predict circRNAs using either CIRI2 [26] or STAR Chimeric Post (STARChip) [27] starting from RNA-Seq reads.

The CIRI2 prediction analysis is implemented by the function *wrapperCiri*. This function initially calls the *fastqc* function for quality control of the input RNA-Seq reads using FASTQC (Available online: http://www.bioinformatics.babraham.ac.uk/projects/fastqc). Then it executes the *bwa* function for performing read alignment using BWA [28]. Finally, *ciri2* function is called to execute the CIRI2 algorithm on the read alignment obtained by BWA.

The STARChip prediction-pipeline is implemented through *wrapperSTARChip* embedding the functions *starChimeric*, *starChipIndex*, and *starchipCircle*. Before running *wrapperSTARChIP*, the genomic sequence should be indexed using STAR through the function *rsemstarIndex*. In *wrapperSTARChIP* the RNA-Seq reads are aligned against indexed genomes using the *starChimeric* function which exploits the chimeric alignment mode of STAR algorithm. The chimeric alignments are then evaluated for candidate circRNA junctions using the function *starChipIndex* to pre-process the reference STAR genome index. Finally, the *starchipCircle* function returns a list of circRNAs supported by a user-defined minimum number of BS-supporting reads detected in a defined number of samples. An additional filter based on the count per million reads, the linear splicing information, as well as the circRNA annotation, can be also provided by the function.

Finally, we provided two functions for processing and overlapping multiple circRNA predictions. Specifically, we designed the function *circrnaMergePredictions* to merge the predicted circRNAs in each sample given a specific prediction tool. This function searches within the sample folders the circRNA predictions as stored in files named with a suffix “tool.crna” and “tool.count_table”, where *tool* is the chosen prediction tool (e.g., suffix *ciri2* for CIRI2 prediction). Then, it creates a joined BS count table from different samples and removes those circRNAs characterized by a few BS-supporting reads in each sample and by a low average number of BS-supporting reads among biological replicates of the same experimental condition (these two thresholds are passed to the function *circrnaMergePredictions* as input parameters). Finally, the function *circrnaOverlapResults* was designed to merge circRNAs predictions derived by different tools. It returns a list of predictions in which only the circRNAs discovered by at least N tools are considered (where N is an input parameter of this function). It is important to point out that these two functions are currently implemented to support the circRNA predictions derived from eleven tools: ACFS [28], CIRI [29], CIRI2 [26], Find_Circ2, CIRCexplorer [30], CIRCexplorer2 [31], DCC [17], KNIFE [32], STARChip [27], Uroborus [33], and circRNA_Finder [34].

#### 2.1.2. Module 2: circRNAs Classification and Annotation

In this module, an extensive annotation of the circRNAs predicted in Module 1 is performed through the following two functions: *circrnaClassification* and *circrnaAnnotations*. The *circrnaClassification* function considers the Ensembl transcriptome annotations by overlapping the exons genomic coordinates against circRNAs genomic coordinates. Each overlap is classified on BS position within the annotations and the number of exons involved. This function is able to distinguish unconventional circRNAs classes including intronic and intergenic circRNAs, or circRNAs whose BS sites do not coincide with exon boundaries (putative exon circRNAs), as reported in [19]. Specifically, if both circRNA BS sites coincide with the exon boundaries then circRNAs are classified as monoexonic or multiexonic depending on the number of exons involved. Conversely, the circRNAs are classified as intronic, intergenic, or putative exon if at least one BS site falls within intronic, intergenic, or intra-exonic regions, respectively. The *circrnaClassification* function takes as input the list of predicted circRNAs, the exons/transcripts data, and the selected genome assembly. The output of this function is thus a circRNAs classification at the transcript and gene level. The *circrnaPrepareFiles* function can be used to retry the exons/transcript data, in case they are not available, using the biomaRt R package [35]. These functions can be applied on circRNA sets defined in human transcriptomic data (hg18, hg19, or hg38 genome assembly) as well as on data obtained in model organisms, including *Mus musculus* (mm9, mm10 assembly), *Rattus norvegicus* (Rn6), Drosophila melanogaster (dm6), and Caenorhabditis elegans (ce11). 

On the other side, the *circrnaAnnotations* function compares the list of predicted circRNAs with a set of online circRNA databases. Based on the genomic coordinates of the input circRNAs, the function provides to the user their related information by querying six databases: circBase [5] and TSCD [6], ExoRBase [9], Circ2Disease [8], CSCD v2 [10], and CircFunBase [11]. Each circRNA identified in these databases is associated with their annotations including: Genomic coordinates and length, the cell lines in which a circRNA was detected, the best overlapping gene and transcript, the study in which it was discovered (from circBase), the information about the fetal or adult human and mouse tissues in which the circRNA was detected (from TSCD), the circRNA expression in disease and normal tissues (from CSCD, Circ2Disease, CircFunBase), the circRNA detection in circulating exosomes (from ExoRBase), andmiRNAs and proteins predicted to bind its sequence (from TSCD). 

Noteworthy, if the version of the genome assembly used for the circRNA prediction is not compatible with those present in circBase and TSCD, the genomic coordinates are converted using the UCSC LiftOver program [36]. The *circrnaAnnotations* is compatible with the human (hg18, hg19, and h38) and mouse (mm9, mm10) genome assemblies.

#### 2.1.3. Module 3: circRNAs Sequence Analysis

This module is designed for the analysis of the back-splicing circRNAs sequence. The *circrnaBSJunctions* function takes as input the list of selected circRNAs and the human genome sequence providing the reconstructed BS sequences of the input circRNAs. The function exploits a python script which identifies two sets of genomic coordinates of 35 base pairs starting from the boundaries of the exons involved in the circularization. To reconstruct BS sequences, these genomic coordinates are then used as input for the functions *getSeq* and xscat provided by the R package GenomicRanges [37]. *circrnaBSJunction* can be applied on circRNA set defined in human data (hg18, hg19, and hg38) or defined using model organisms data (mm9, mm10, Rn6, dm6, and ce11).

Moreover, *ciri_as* function implements the CIRI-AS analysis to detect internal alternative splicing (AS) events involving the exons composing the predicted multi-exonic circRNAs [15]. Specifically, *ciri_as* takes as input BWA alignment files used for the CIRI2 circRNAs prediction, the reference genomic sequence (in fasta format), the gene annotations (in GTF/GFF), and the list of CIRI2-predicted circRNAs. Its output is the list of alternative splicing events involving the exons of the circRNAs under study.

#### 2.1.4. Module 4: circRNAs Expression Analysis

This module provides the expression analysis of circRNAs. The differential expression analysis is based on the DESeq2 R package [38]. The function *wrapperDeseq2* takes as input the circRNA BS count table generated by the R functions provided by the Module 1 and computes the differential expression analysis for each circRNA with respect to a specific covariate.

Conversely, if the user is interested in quantifying the expression of the set of circRNAs predicted in the previous modules in other independent RNA-Seq dataset, we provided the function *circrnaQuantification*. This function applies a two-step procedure to count the sequencing reads supporting a BS junction. In practice, the first step of the function takes as input the sample reads, the set of sequences and a threshold *N* and it returns the corresponding set of reads which contains at least *N* sub-sequences of length *k* (called *k*-mer) shared with the set of pre-defined BS sequences. The *k*-mers are stored in RAM exploiting an ad-hoc C++ hash table class implementation to optimize the trade-offs between the memory utilization and the execution time.

The second step takes as input the reads selected in the first step and directly align them against the pre-defined BS sequences. For this step, the Smith–Waterman algorithm provided by SIMD Smith–Waterman C++ library [39] is used. The *circrnaQuantification* function can be applied by providing the input circRNAs BS junctions in fasta format, the RNA-Seq data to analyze in fastq format, and six parameters: the *k*-mer length, the number of threads, the dimension of the hash table, the dimension of the collision list, the number of *k*-mers that must be matched to the sequence and the number of perfect matches required to consider the sequence represented in the RNA-Seq data. The circRNAs BS count table obtained can then be joined together using the *mergeData* function and analyzed by the *wrapperDeseq2* function for a differential expression analysis.

#### 2.1.5. How to Integrate a New Functionality in the Framework

As part of the RBP, Docker4Circ allows also users to include novel functionality in the framework by creating their own functions and Docker images. For this purpose, a template function called *skeleton* is provided as a prototype to build the docker controlling function. Furthermore, a tutorial on “how to create the docker image called via the skeleton.R function” was created (see available online: http://www.reproducible-bioinformatics.org/ in the section “How to be part of the Reproducible Bioinformatics project”).

### 2.2. Examples of the Application of Docker4Circ Framework

In the following section, we used Docker4Circ to identify the set of circRNAs expressed from RNA-Seq data of normal colon mucosa (NCM) NCM460 cell line and colorectal cancer (CRC) SW620 and SW480 cell lines [40] (Section 2.2.1). The expression analysis module was also used to quantify the expression of the circRNAs identified in an RNA-Seq dataset of primary CRC and paired NCM tissue data set (GSE104178) [41]

#### 2.2.1. Docker4Circ for the Reproducible Analysis of circRNAs Expressed in Colorectal Cancer Cell Lines

The Docker4Circ modules were used to predict, classify, annotate, and analyze the expression of the circRNAs in NCM and CRC cell lines. The expression analysis (Module 4) was exploited to measure the expression of the detected circRNAs in an RNA-Seq experiment obtained from primary CRC tissues and adjacent normal tissue.

The prediction of circRNAs in NCM and CRC cell lines was performed using the CIRI2 pipeline, implemented by the *wrapperCiri* function. As reported in Table 1, among the different samples the lowest number of circRNAs predicted by CIRI2 was 4,624, while the highest circRNAs predicted was 16,006 (average value = 9,474). The complete list of circRNAs, together with the number of BS-supporting reads for each sample, is reported in Appendix A. The number of circRNAs decreases from NCM to CRC cell lines, as previously reported by the authors [40].

Using the *ciri2MergePredictions* function we merged the circRNAs predicted in each sample into a single circRNAs list. This list is composed of 7,086 out of 31,694 circRNAs characterized by more than two BS-supporting reads in at least two replicates and an average value of BS-supporting reads higher than 10 (Appendix A). Observe that 99.80% (*n* = 7,072) of the circRNAs belonging to our list was previously detected by Jiang et al [40]. Starting from the same datasets, the circRNAs prediction was also performed with the STARChip pipeline. The analysis predicted 2,933 circRNAs of which 94.7% overlapped with CIRI2 (Appendix A).

As previously observed by our group, circRNAs can be synthesized by complex splicing patterns involving exonic, intronic, and intergenic regions [19]. To better characterize the genomic regions involved in the back-splicing process of the 7,086 circRNAs predicted by CIRI2, we applied the function *circrnaClassification* that provides an accurate classification of each circRNA based on BS site location with respect to Ensembl transcript annotations. At the transcript level, the exons of 15,454 transcripts were associated with at least one circRNA, while 7,200 unique classifications were provided at the gene level (Figure 2b). The discrepancy between the number of inputs circRNAs and the unique classification results is due to those circRNAs that can be attributed equally to two or more genes sharing the same exons. Appendix A provides the classification results at the gene level with information of involved exons. Given our circRNAs classification, we observed that exonic circRNAs were the class associated with the highest expression and the circRNAs predicted on the *HIPK3* (exon 2), *CAMSAP1* (exons 2–3), and *ASXL1* (exons 2–4) were the most expressed circRNAs in both normal and cancer cell lines (Appendix A). These circRNAs were also identified by Jiang and coworkers as the most expressed circRNAs [40].

To further characterize the structural properties of the circRNAs in our list, we applied the sequence analysis module of Docker4Circ to identify AS events involving the exons composing the circRNAs. This analysis was performed with CIRI-AS algorithm implemented in the *ciri_as* function and generated an average of 780.3 AS events involving multi-exonic circRNAs in our set (Table 1 and Appendix A).

Subsequently, to describe our circRNA set based on public database information, we applied the *circrnaAnnotations* function of Docker4Circ. The annotation with data from the six databases highlighted 4,950 circRNAs annotated in CircBase, 540 and 508 circRNAs annotated in the adult and fetal tissue sections of TSCD database, 334 in CircFunBase database, 24 in ExoRBase, 72 in Circ2Disease, and 29 in CSCD v2, respectively (Appendix A). Interestingly, 83 circRNAs were detected in adult colon tissue samples in TSCD data and 26, 2, and 29 were detected in CRC tissue using the CircFunBase, Circ2Disease, and CSCD v2 annotations, respectively (Appendix A). A total of 1,132 circRNAs were not annotated in any of the considered database.

Finally, using the functions implemented in the expression analysis module, we assessed the differential expression level of our list of circRNAs among different experimental conditions or from independent RNA-Seq experiments. Then, we performed a differential expression analysis (*wrapperDeseq2* function) considering the number of BS-supporting reads measured in NCM and CRC cell lines. As reported in Appendix A, we identified 705, 655, and 430 circRNAs differentially expressed (adj. *p*-value < 0.001) between NCM460 and SW480 cell lines, NCM460 and SW620 cell lines, and SW480 and SW620 cell lines (Appendix A). Among them, 639 (90.64%), 613 (93.59%), and 352 (81.86%) were detected as differentially expressed also by Jiang and coworker [40]. Finally, 208 circRNAs were significantly differentially expressed in all the comparisons. Among them, the circRNAs mapped in *EXOC6B* (exons 1–3), *DCBLD2* (exons 1–2), and *ASAP1* (exon 2) were the most significant dysregulated circRNAs (Appendix A).

#### 2.2.2. Application of Docker4Circ to Directly Quantify circRNAs Expression from CRC Tissue RNA-Seq Data

The expression of the 7086 circRNAs identified using the cell lines data described in the previous section, was quantified in an RNA-Seq dataset of primary CRC and paired NCM tissue data set (GSE104178) [41]. For this purpose, the sequences of the circRNAs back-splicing junctions were reconstructed using the Docker4Circ *circrnaBSJunctions* function. Using the hash table-based approach, the reconstructed BS sequences were searched directly in total RNA-Seq reads bypassing the circRNAs prediction in each tissue sample. Using this quantification method, 1,758 circRNAs were associated with at least one read. The most expressed circRNAs in NCM samples was chr17_45043900_45047675 an intronic circRNA of *RP11-156P1.2* gene, whereas a circRNA from the *RPA3-AS1* gene (exons 2 and 3) was the most expressed circRNA in CRC samples.

Then, the BS read count table was used as input of a differential expression analysis (*wrapperDeseq2* function) between the CRC and the paired NCM datasets. We identified six circRNAs differentially expressed between CRC and normal colonic mucosa samples (*p*-value < 0.01) (Figure 2c and Appendix A). Among the six differentially expressed circRNAs between primary tumors and matched normal mucosa, two circRNAs were detected as differentially expressed using the CRC cell line data. Specifically, a circRNA encoded by the *PSMA3* (chr14_58718837_58724716) and one encoded by the *HDAC2* gene (chr6_114274441_114277315) were upregulated in primary tumors. The circRNA chr14_58718837_58724716 was significantly up-regulated in SW480 compared to NCM460 cell lines, while chr6_114274441_114277315 was up-regulated in SW640 compared to SW480 cells.

## 3. Discussion

CircRNAs are widely expressed in both cancerous and normal tissues [42,43] and an increased number of sequencing experiments is becoming accessible to explore circRNAs expression in a specific biological context. To deal with the increasing number of computational resources and public datasets available for circRNAs analysis, we propose Docker4Circ as a user-friendly framework to guarantee reproducible analysis of circRNAs data.

The framework was designed for users with different levels of expertise in computational analysis. Specifically, a Java graphic user interface was designed to provide a basic user-friendly framework. Conversely, more expert users can exploit the R environment to create their own analysis workflows using directly the R functions implemented in Docker4Circ.

The four modules composing the Docker4Circ framework facilitate the circRNAs prediction starting from raw RNA-Seq reads and the comparison of multiple circRNA lists (Module 1), the circRNA characterization by defining their genome context and the annotated knowledge about their expression (Module 2), the reconstruction of the circRNA BS sequence to easily design qPCR primers for validating circRNAs expression (Module 3), and the rapid quantification and differential expression analysis of circRNAs level using public datasets (Module 4). Indeed, starting from RNA-seq datasets, Docker4Circ provides all the features and annotations needed for further experimental work, such as the BS sequence, the genomic status and contextualization of the circRNAs within other linear cognate transcripts and alternative splicing isoforms, and potentially interacting RBPs and miRNAs. This is a consistent advantage with not having to jump to different packages for each of these aspects. In addition, along with the differential expression analysis run inside the original dataset, the fourth Docker4Circ Module allows the circRNAs identified in the original dataset to be quite rapidly analyzed in other independent external datasets thanks to the function *circrnaQuantification*.

We tested Docker4Circ to reproduce analyses performed by Jiang and colleagues [40] on circRNAs expression in CRC cell lines showing that our framework is able to reproduce extensively their results. Furthermore, we added novel evidence on these circRNAs by executing a quantification and differential analysis of their expression level considering RNA-Seq performed on CRC and adjacent colonic tissues. This analysis showed two circRNAs differentially expressed both in tissues and cell lines models (Appendix A). These circRNAs were annotated, respectively to the *HNRNPC* (exons 1–2) and the *PSMA3* (exons 3–5) genes. Furthermore, considering the circRNAs detected as differentially expressed in CRC cell lines data, we identified 355 circRNAs detected in the primary tissue datasets and five of them are annotated to CRC considering the Circ2Disease or CircFunBase annotations. Specifically, the circRNA hsa_circ_0005273 (from the *PTK2* gene) was annotated to CRC disease in both circ2disease and CircFunBase, while hsa_circ_0002321, hsa_circ_0005576, hsa_circ_0004820, and circ_004661 transcribed respectively from the *PPP2R5A*, *CDC42*, *NUP35*, and *PTPRA* genes were annotated to CRC in CircFunBase. Despite further experimental validations are needed to assess the expression of these circRNAs in CRC and normal colonic mucosa, we showed that our approach is able to provide a reproducible prediction and characterization of a circRNAs set.

As reported in Appendix A, all the modules and functions implemented in Docker4Circ can be run in a limited amount of time and the overall running time of the workflow was around six or ten hours if the CIRI2 or the STARchip prediction is performed, respectively. The workflow was performed on an Intel NUC6I7KYK mini-PC with 8 threads confirming that all the Docker4Circ functions can be executed on a standard workstation because the only requirement is 32 Gb of RAM available if the STAR Chimeric analysis is performed. Moreover, one additional aspect that should be considered is the time and computational power requested to run circRNA analysis from RNA-Seq data concerning elevated number of samples, such as in the case of tumor tissue or sera series, often involving (several) hundreds of samples. For this purpose, we included a hash-based circRNA expression quantification (*circrnaQuantificiation* function) which allows the direct computation of circRNA expression level from the sequencing reads avoiding the read alignment step, limited to a pre-defined set of circRNAs.

The most important aspect of our framework is its ability to guarantee the computational reproducibility of the analysis. This is obtained by embedding each analysis step into a specific docker image according to the guidelines of the RBP project. Thanks to this, our framework provided analyses whose results can be fully tracked on how they were produced by recording each analysis steps and version of tools applied without any data manipulation step.

In conclusion, Docker4Circ provides an efficient analysis framework to identify and characterize the circRNAs on a large number of sequencing experiments. The usage of Docker images ensures a reproducible circRNAs analyses to easily harmonize and combine the study of these molecules in different experimental and biological contexts.

## 4. Materials and Methods

The detailed analysis protocol that was followed for the analyses reported in this manuscript was released on the protocol.io portal at dx.doi.org/10.17504/protocols.io.9vmh646. The specific use of each function and associated parameters are reported in Appendix A of the manuscript.

### 4.1. CircRNAs Prediction

To test Docker4Circ, RNase-R RNA-Seq datasets from PRJNA393626 were selected. These data consist of a triplicate paired-end total and RNase-R treated RNA-Seq performed on NCM460 (normal colon cells), SW480 (primary CRC cells), and SW620 cell lines (metastatic CRC cells).

CircRNAs prediction was performed using the *wrapperCiri* function of Docker4Circ with the following parameters: max.span = 200,000, stringency.value = “high”, and quality.threshold = 10. Using this function, reads alignment was performed using the mem mode of BWA v.0.6.1 in default settings. CircRNAs prediction was performed using CIRI2 algorithm v.2.06 [26]. Gencode v28 was used as reference transcriptome while Ensembl hg19 (GRCh37) as Human reference genome. CircRNAs predicted in at least two out of the three biological replicates in each condition and associated with an average number of BS-supporting reads >10 were selected. This prediction overlap was performed with the *ciri2MergePredictions* function of Docker4Circ using the options min_reads = 2, min_reps = 2, and min_avg = 10. The list of circRNAs was overlapped with those predicted in [40] by converting the circRNA genomic coordinates from hg19 to hg38 human genome assembly using LiftOver algorithm [36].

For the circRNAs prediction with the STARChip pipeline, the reference genome was indexed using the function *rsemstarIndex* and *starChipIndex*. Subsequently chimeric read alignments for each dataset were detected using the function *starChimeric* with parameters chimSegmentMin = 20 and chimJunctionOverhangMin = 15. Finally, the function *starchipCircle* was applied to predict the circRNAs using the STAR Chimeric alignments. The function was applied with parameters reads.cutoff = 1, min.subject.limit = 2, do.splice = “true”, cpm.cutoff = 0, subjectCPM.cutoff = 0, annotation = “true”. The *ciri2MergePredictions* function was used to filter the circRNAs read count table using the same parameters exploited during the CIRI2 analysis. The overlap between CIRI2 and STARChip circRNA predictions was performed by considering their genomic coordinates.

### 4.2. CircRNAs Classification and Annotations

The circRNAs classification was performed using *circClassification* function (with option assembly = “hg19”) applied on the list of circRNAs predicted by CIRI2 and on the reference hg19 transcript annotations from Ensembl (Ensembl v93) downloaded using the *circrnaPrepareFiles* function of Docker4Circ (with option assembly = “hg19”).

The circRNA annotation was performed using the *circrnaAnnotations* function with option genome.version = “hg19”.

### 4.3. CircRNAs Sequence Analysis

The 70bp sequences representing the reconstructed circRNA BS junctions were obtained using the function *circrnaBSJunctions* of Docker4Circ. The resulting fasta file was used for the quantification analysis. Prediction of internal alternative splicing events involving multi-exonic circRNAs was performed using the *ciri_as* function on each list of circRNAs predicted by CIRI2.

### 4.4. Quantification of circRNAs in RNA-Seq Datasets

The *circrnaQuantification* function of Docker4Circ was applied using six RNA-Seq datasets from GSE104178 [41]. These data consist of total RNA-Seq performed on three matched pairs of colorectal cancer (CRC) samples and matched normal colonic mucosa (NCM) samples. The quantification analysis was performed selecting a k-mer length equal to 21 based on the read length (75 bp); a minimum number of matching k-mer equal to 17, and a minimum number of perfect matches equal to 30. The maximum number of the element stored in the hash table was set to 1,000,003. DESeq2 v1.20.0 [38] was applied for BS read count normalization and differential expression analysis. The algorithm was applied in default settings. The analysis was performed using the *wrapperDeseq2* function of Docker4Circ on the result of the *mergeData* function which was used to join different circRNA count tables with the covariates indicating the samples classes.

### 4.5. Availability of Source Code and Requirements

The Docker4Circ R functions were integrated into the Docker4Seq R package available at https://github.com/kendomaniac/docker4seq.

The Java GUI can be downloaded from https://github.com/mbeccuti/4SeqGUI.

The analysis is independent of the Linux operating system applied, while Docker software is required. All Docker4Circ modules are already integrated into a Docker image. Each docker image tag is then created following rule defined by RBP: Docker image tags are labelled with the extension YYYY.NN, where YYYY is the year of insertion in the stable version and NN a progressive number. YYYY changes only if any update on the program(s), implemented in the docker image, is done. This because any such updates will affect the reproducibility of the workflow. Previous version(s) will be also available in the repository. NN refers to changes in the docker image, which do not affect the reproducibility of the workflow.

The Docker4Circ docker images can be created using the Docker files provided in https://github.com/cursecatcher/biodocker/blob/master/docker4ciri/Dockerfile. 

The Docker4Circ pipeline can be run by the Docker4Seq R package, by the 4SeqGUI Java graphical interface, and directly by the bash command lines using the commands reported in the Appendix A of the manuscript.

### 4.6. Docker4Circ Running Time Estimation

The running time of each analysis was computed by considering the execution using 8 threads of an Intel NUC6I7KYK mini-PC [25]. Specifically, multithreading was exploited only for the execution of the functions *ciri2*, *circrnaQuantification*, *starChimeric*, and *starchipCircle*.

## Figures and Tables

**Figure 1 ijms-21-00293-f001:**
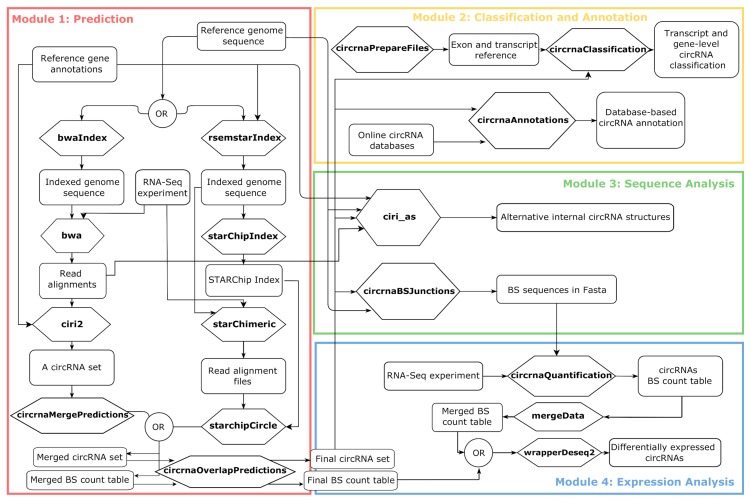
Schematic representation of the Docker4Circ modules with indication of all the functions (reported in bold in the hexagons) and the input/output files involved (reported in the squares). The different modules implemented in the framework are reported with different colors. BS = back-splicing.

**Figure 2 ijms-21-00293-f002:**
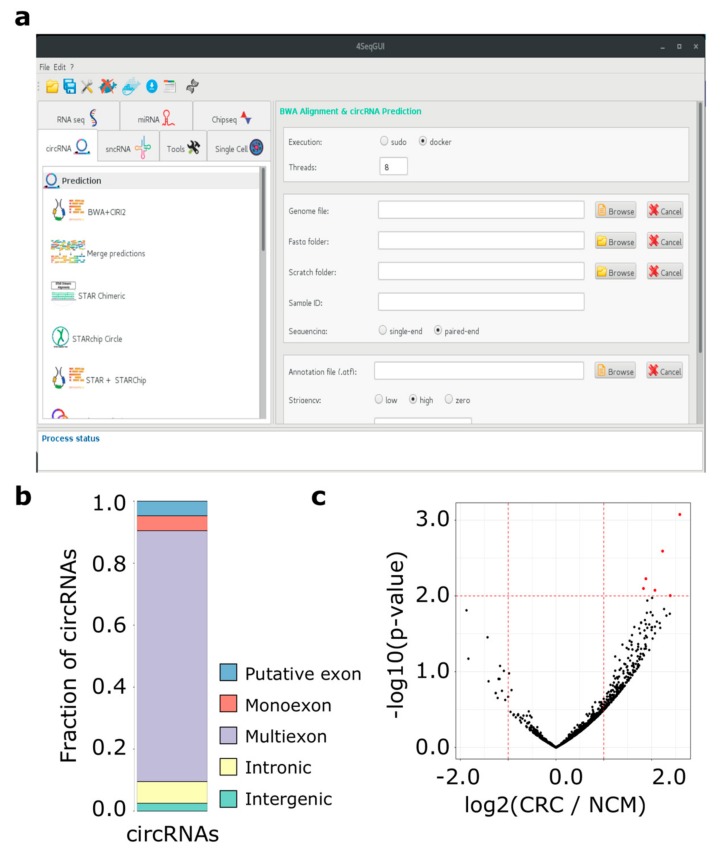
(**a**) The Docker4Circ Graphical User Interface. Each module implemented in the framework is accessible using the panel on the left, the right panel reports the fields and parameters of each function; (**b**) Bar plot reporting the Docker4Circ classification of circRNAs identified in the analysis of RNA-Seq datasets from CRC cell lines; (**c**) Volcano plot reporting the -log10 p-value and the log2 expression fold change (red dashed lines) computed between Docker4Circ counts of BS supporting reads from RNA-Seq datasets of NCM and CRC tissue samples.

**Table 1 ijms-21-00293-t001:** Table reporting the number of RNA-Seq paired reads analyzed, the number of detected circRNAs, and the number of alternative splicing (AS) events predicted by CIRI-AS.

Dataset ID	Reads	circRNAs	AS Events
NCM460_R1	66,144,999	14,003	1,482
NCM460_R2	70,945,094	16,006	1,790
NCM460_R3	73,804,226	12,413	1,078
SW480_R1	88,915,933	8,627	532
SW480_R2	97,303,573	5,688	335
SW480_R3	66,144,999	7,154	470
SW620_R1	91,406,400	1,0216	790
SW620_R2	67,013,355	4,624	214
SW620_R3	69,789,394	6,541	332
Average	76,829,774.78	9,474.67	780.33

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
