# Peer review of "Docker4Circ: A Framework for the Reproducible Characterization of circRNAs from RNA-Seq Data"

_ijms, 2019, doi:10.3390/ijms21010293_

Round 1

Reviewer 1 Report

Authors replayed to all my comments satisfactorily

Reviewer 2 Report

The revised manuscript has been improved accordingly.

This manuscript is a resubmission of an earlier submission. The following is a list of the peer review reports and author responses from that submission.

Round 1

Reviewer 1 Report

The article presents a toolbox for characterising the circRNAs from RNAseq data through an easier to use package and user interface, while insuring reproducibility of the analysis. This type of packaging is generally useful and similar initiative turned out to be very successful and popular in related fiels, for example Qiime2. The present implementation however has many shortcomings, many I believe due to the authors trying to egoistically push forward the results of their own research rather than to provide a tool that is truly of universal usefulness. First, the authors makes a strongly opinionated choice on the tools to integrate to their pipeline and do not offer any flexibility to the user to select an alternative method. This is unlike, once again, the rather successful Qiime2 package where the developers are simply integrating tools from the recent research in an easy to use package while remaining neutral on which tool to user. Secondly, the fact of using Docker containers by itself does not guarantee in any ways reproducibility of the results. Containers are a helpful tool to achieve reproducibility, but the way the containers are designed and built is key for that. In here, the authors provides binary containers on a public repository hosted by a private organisation ( https://hub.docker.com/r/repbioinfo/docker4circ.2019.01 ). In this framework, the end user needs to base its research efforts on the trust that these containers won't be modified, taken down, hacked, etc... At the minimum, the authors should make the DockerFile for these containers clearly available together with their publication and the reviewers should verify wether these can be built reproducibly. Third, the expectations for a docker application in the docker community is to be able to run applications with no other dependencies than Docker itself. In here, the authors are running containers through an R interface, forcing the user to install R itself (and probably a specific version of it) on the system. I would thus strongly recommend the authors to package their entire application in a single docker container so that the entire system can installed and run using a single "docker pull/docker run" command. Overall, given the shortcomings above, I do not see that this tool in its current state can be reasonably used outside of a small community close to the authors themselves. I thus recommend against publication of this article. A significant amount of work and a change of mindset is needed to make it more universally useful.

Reviewer 2 Report

The manuscript proposed by Ferrero and colleagues describes a new algorithm to identify circRNAs and to evaluate their expression from RNA sequencing experiments. The manuscript is well organized but it is lacking of validations based on molecular biology, also considering that authors are not all bioinformaticiansand that the aims of the journal are “provides an advanced forum for molecular studies in biology and chemistry, with a strong emphasis on molecular biology and molecular medicine.” Below some suggestions to improve the manuscript and to allow its publication in this Journal.

Major comments

1. I suggest to verify if differentially expressed circRNAs (at least 5 since this is a genomic based approach) are really present in CRC cell lines or tumor samples.
2. I suggest to verify the expression of validated circRNAs. Are they really differentially expressed between normal and tumor samples?
3. Please provide a reason for using CIRI2 and STAR Chimeric Post algorithms in the proposed workflow. Many other algorithms to predict circRNAs are available (ACFS, CICRexplorer, CIRCexplorer2, circRNA_finder and so on).
4. Please evaluate if the chosen algorithms work in the same way using the same dataset. Overimposition of predictions and final results.
5. Is it possible to combine results from the two algorithms for further analyses or it is necessary to re-analyze twice the dataset?
6. Why the Authors used these two databases (circbase and TSCD)? Please provide a reason.
7. On the line 147 Authors say: “the list of selected circRNAs and the human genome sequence providing…” Is this software working only on human sequences? It would be interesting to implement it on other genomes especially for model organisms used to study different pathologies (e.g. Mouse, Zebrafish, D. Melanogaster). Please include this possibility and specify it in the text.
8. Please provide an explanation of STARChip result in comparison to CIRI2. With CIRI2 Authors identified 7,072 circRNAs and lesser than half using STARChip.
9. This reviewer does not understand if more expressed and more differentially expressed circRNAs in cell lines were also differentially expressed in tissue data set. Please specify this in the text and comment it.

Minor comments

1. Line 43 arising instead of arise.
2. Line 67 algorithms instead of algorithm.
3. Define BS in the line 80.

Reviewer 3 Report

In this manuscript, the author constructed a circRNA workflow for ease of use and reproducibility in RNA-seq data analysis. I have several concerns.

More data should be analyzed by the workflow to assess applicability. Compare the analysis results with experimental data to ensure the reliability.